# Design and Modelling a Graduated Dispenser for Metabolic Diseases—Phenylketonuria

**Corina Adriana Dobocan** [1,*] **, Emanuela Pop** [1] **, Monica Bogdan** [2] **and Catalin Grec** [1]

1 Department of Design Engineering and Robotics, Technical University of Cluj-Napoca, 400020 Cluj-Napoca, Romania
2 Department of Management and Economical Engineering, Technical University of Cluj-Napoca, 400641 Cluj-Napoca, Romania
* Correspondence: corina.dobocan@muri.utcluj.ro

**Abstract:** In metabolic diseases such as phenylketonuria (a rare disease), a very important way to keep the patient healthy is the administration of amino acid substitutes. This dispenser was designed because in other places (except home) for patients, it is very difficult to take the substitute powder due to the custom weight, which depends on the body weight. We designed and made on a 3D printer a graduated dosing device (12 g) which can be used very easily and has all the elements to be transported. In this way, the necessary dose of amino acid substitutes can be administered to patients with phenylketonuria, including infants aged 6 months–1 year, at the kindergarten or any other places with the absence of a food scale. This dispenser is very easy to carry and very useful to patients.

**Keywords:** graduated dispenser; design; modelling; 3D printing; metabolic disorder

## 1. Introduction

Phenylketonuria (PKU) is a genetic metabolic disease caused by a deficiency of the enzyme phenylalanine hydroxylase, which transforms the essential amino acid phenylalanine (Phe) into tyrosine (Tyr) through the hydroxylation system. In phenylketonuria, tyrosine thus becomes an essential amino acid [1]. This system has as basic components: the enzyme phenylalanine hydroxylase (PAH), tetrahydrobiopterin (BH4) and other enzymes [2].

In phenylketonuria, the treatment is a protein-restricted diet that should be initiated by a dietitian specialized in this field. In our country, the children's parents are the ones who learn how to design the diet, the allowed foods, the forbidden foods and how to administer the amino acid formula according to the instructions of the responsible doctor from the Regional Center. In this way, due to the situation that requires it, the mother of the child/children learns how to calculate the amount of phenylalanine that the child can tolerate depending on the resulting plasma phenylalanine value.

As tools for observing the diet, it is mandatory to purchase a kitchen scale (sensitive to weight), a pocket scale, calculator, and a register in which everything the child eats is written down for each meal. There are also some applications for the calculation of phenylalanine, but accessibility and understanding is conditioned by understanding the names of the products from the database in English and German. It is therefore necessary to create an application in Romanian so that it is accessible to all parents.

If dietary intervention is not taken in the first 20 days of life, high concentrations of phenylalanine accumulate in the blood and cross the blood–brain barrier, causing serious and irreversible damage from a neurological point of view. So, the treatment should start as early as possible, preferably before 20 days of age. Dietary and nutritional interventions should be initiated when the plasma phenylalanine value exceeds 360 µmol/L, values which are detected through the neonatal screening funded by the National Screening Program for Phenylketonuria and Hypothyroidism.

To maintain the value within the safe range (120–480 μmol/L or 2–6 mg/dL), it is necessary to observe the diet, the consumption of low medical foods with reduced phenylalanine content and the administration of the amino acid formula. Dosage of the amino acid formula does not depend on individual tolerance, but on the protein requirements of the PKU subject. To ensure the absorption of these amino acids from the formula, it is recommended that the protein requirement be increased by 40% compared to the recommendations for healthy people. The amino acid formula has been fortified with tyrosine, but it does not always cover the requirements for this amino acid, especially in adolescents and adults. Because the diet is restrictive, vitamins, minerals and carbohydrates are also found in these amino acid formulas [3].

In addition to monitoring the level of plasma phenylalanine, other nutritional indices must be evaluated annually, such as: plasma amino acids, plasma homocysteine or methylmalonic acid (indicators of vitamin B12 deficiency), hemoglobin and ferritin. The dosing of vitamins and minerals is also recommended, the most important being Ca, Zn, Se and the hormonal blood dosage. Growth parameters are also monitored, following the child's evolution, and neurological and neuropsychiatric tests are recommended if there are suspicions of neurological damage [4,5].

Low protein foods with reduced phenylalanine content ensure an intake of 85% of the protein requirement, the difference of 15% being provided by natural proteins (fruits and vegetables). These medical foods are indispensable for PKU patients because they prevent protein deficiency, ensure energy requirements, metabolic control, and normal growth, and prevent zinc, iron, carnitine, selenium and vitamin B12 deficiency [6,7]. Low protein foods are: flour, bread, pasta, cereals, pizza, rice, egg substitutes, meat substitutes, purée substitutes, and mixes for pastries, pancakes, muffins and other assortments.

The worst neurological manifestations of this metabolic disease are prevented by early diagnosis, establishing and following the specific therapeutic diet. Even if there is adequate control of plasma phenylalanine, some behavioral, physical and neuropsychological problems may occur. Recent studies have shown that working memory alterations occur in PKU with a neonatal diagnosis since it has been associated with low adherence to treatment.

The care of people with PKU requires a multidisciplinary effort (specialist doctor, dietitian, psychologist) and support is needed throughout life to maintain a good result, which consists in a normal development. Even if phenylalanine levels are within the safe range, changes in body composition may occur. Several studies in the field have shown that the studied patients with phenylketonuria show an increase in the percentage of energy due to carbohydrates compared to the percentage of energy from fats and proteins. Because subjects must restrict protein intake, this causes fat intake to decrease, and energy requirements are met by excess carbohydrate consumption [6].

The administration of the amino acid substitute requires the possession of a scale (either food or pocket) because it must be weighed according to the necessary doses, and it is very important to have the availability of administering this substitute in any place: e.g., kindergarten, school, city or other places different from the normal environment. This device greatly facilitates the possibility of administration and is easy to use.

The design of this dispenser was carried out in the design laboratories of the Technical University of Cluj-Napoca, Faculty of Industrial Engineering, Robotics and Production Management, by the team at the Department of Design Engineering and Robotics.

## 2. Materials and Methods

The materials used were Autodesk Inventor design software and FFF (Fused Filament Fabrication) 3D printing technology. The printer used was the Prusa i3 MK3S+.

It is a method of manufacturing real objects in the form of prototypes that is initiated by the existence of an idea. The choice of printer and its calibration and configuration settings as well as the quality of the plastic filament used [8] are important [9,10].

The main printing technologies used are the following:

FDM—Fused Deposition Modeling;

SLA—Stereolithography;
DLP—Digital Light Processing;
SLS—Selective Laser Sintering;
SLM—Selective Laser Melting/Direct Metal Laser Sintering;
3DP—Tridimensional Printing inkjet;
LOM—Laminated Object Manufacturing;
PJP—PolyJet Printing.

Rapid prototyping technology FDM (Fused Deposition Modeling) is presented in Figure 1 and is the most used because of its accessibility and simplicity. It is used in modelling, prototyping and in production applications. This technology also includes MEM (Melting Extrusion Modeling), TPE (Thermoplastic Extrusion) and FFF (Fused Filament Fabrication) [11]. We chose the FDM method because our scope was to make a physical prototype in our CAD lab with the Prusa i3 MK3S+ printer. Of course, other methods could print a much smoother and more robust prototype.

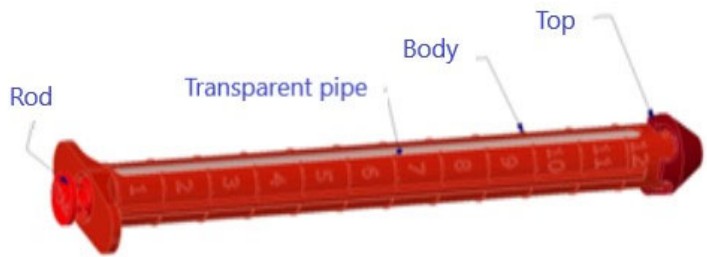

**Figure 1.** The component elements of the dispenser.

With a dedicated software application, the desired 3D model was initially sliced into cross sections called layers. The printing technology consists of passing a filament of plastic material through an extruder that heats it up to melting point, then applying it uniformly (by extruding) layer upon layer with great accuracy to physically print the 3D model according to the CAD file.

The head (extruder) was heated to melt the plastic filament, moving both horizontally and vertically under the coordination of a numerical control mechanism, controlled directly by the printer's CAM application. As it moved, the head deposited a thin string of extruded plastic that upon cooling hardened immediately, sticking to the previous layer to form the desired 3D model.

To prevent deformation of parts caused by the sudden cooling of the plastic, some professional 3D printer models include a closed construction chamber, heated to a high temperature. For complex geometries or cantilever models, FDM technology requires printing with support material that will later have to be manually removed.

## 3. Results

The first step was to design this object (digital modeling) and then generate a file (usually in STL format) that contains all the technical geometrical information needed to export and convert this model into a list of instructions that the 3D printer can execute. According to these instructions, the process of creating the object began by using the 3D technology called FFT (Fused Filament Fabrication). This technology was also used to make this graduated dispenser and consists in modeling the object by melting a plastic filament and overlapping several layers of material. After the printing process was completed, the object was detached from the printing platform, the extra parts were removed, and the object was finished [11].

The CAD design of the medicine tank was done in the Autodesk Inventor program and is presented in Figure 2. The device is composed of eight parts and two gaskets used to seal the powder in the dispenser. The component parts are:

- dispenser body;

- transparent tube;
- tip;
- rod;
- two O-rings;
- two caps;
- a powder loading funnel in the dispenser body.

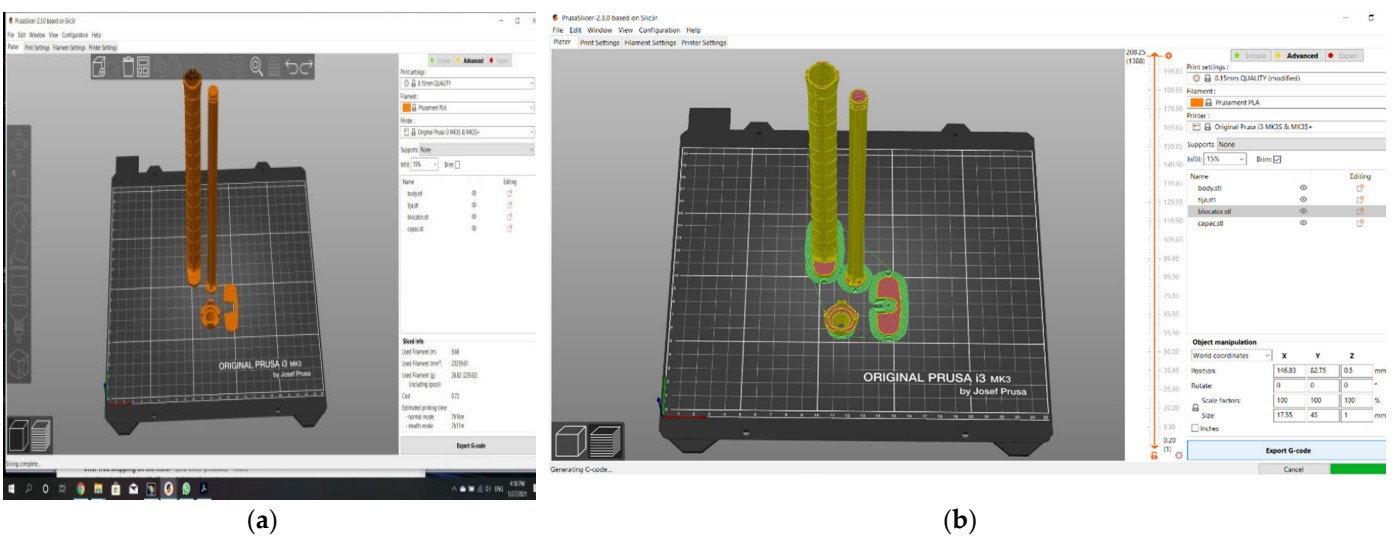

|        |        |
|:------:|:------:|
| (**a**) | (**b**) |

**Figure 2.** The modelling and making process of the dispenser using the Prusa I3 MK3S+ Printer: (**a**) stage 1 and (**b**) stage 2.

The dispenser is graduated and can dose 12 g of substitute powder (12 g $\times$ 0.6 = 7.2 g protein). The dispenser body, tip, rod, caps, and funnel were made on the 3D printer, and the clear tube and gaskets were purchased commercially.

Obtaining the component parts through 3D printing (prototype) using the Prusa I3 MK3S+ Printer (Figure 2), the working time related to the 3D printing of the dispenser was 12 h, the long time being due to printing with a support to achieve a quality as high as possible.

This dispenser was designed for children (aged 6 months–1 year) and made for a protein substitute powder because it is easier to administer if they are in a different place than home. This substitute powder is stored in 500 g drums and this quantity is not always available to be transported. The substitute is administered only enteral, being in the form of powder that dissolves in water, syrups and fruit juices.

In the following, we present the physical achievement of this dispenser in our CAD laboratory (Figure 3). The dimensions of the dispenser (Figure 4) are suitable for a bag and a food box and after use it can be washed easily.

The dispenser dimensions (Figure 4) are suitable for a bag and a food box and after use it can be washed easily. The dispenser is pre-filled with protein substitute powder and can be transported this way because it has two caps that prevent the flow of the powder during transport, as can be seen in Figure 5.

In phenylketonuria, L-amino acids from protein substitutes without phenylalanine do more than provide replacement proteins for growth and to maintain muscle mass, having a role in controlling blood phenylalanine and inhibiting its transport to the brain and absorption in the gut. Evidence suggests that a higher intake of formula increases satiety and thus favors better adherence to treatment and avoids transgressions [3,12–14].

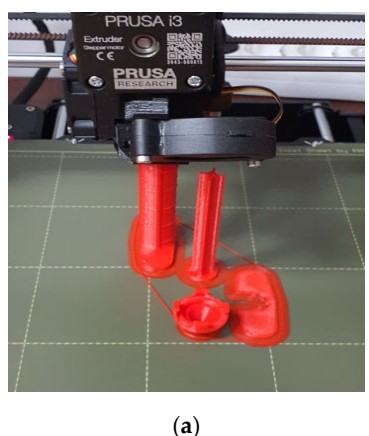 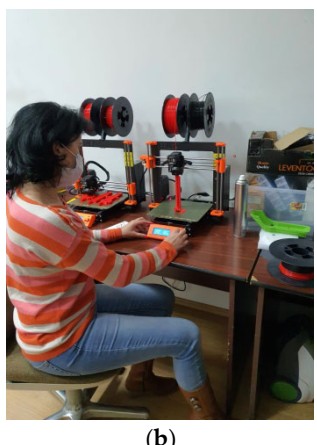

(**a**)            (**b**)

**Figure 3.** The physical achievement of this dispenser: (**a**) Prusa I3 MK3S+ Printer and (**b**) adjusting the printer parameters.

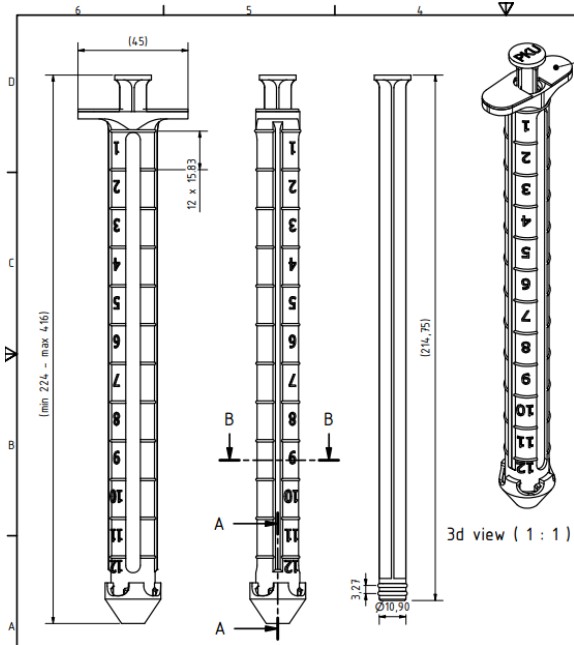

**Figure 4.** The overall drawing of the dispenser.

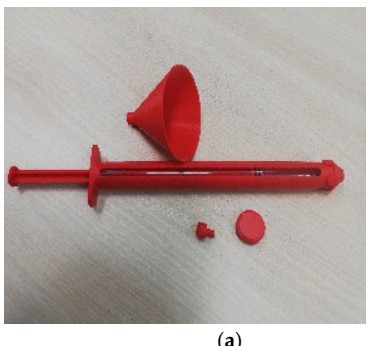 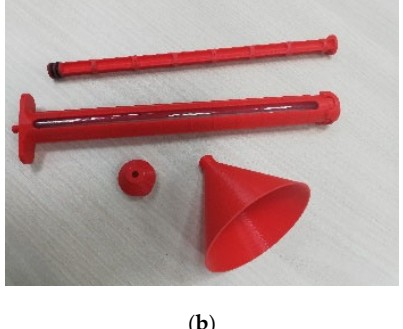

(**a**)            (**b**)

**Figure 5.** The alternative prepared for dosing (**a**) and transport (**b**).

At the time of dosing, the two caps are removed, the rod is fixed and attached to the tip of the dispenser, and the powder is pushed into the prepared container with liquid so

that it can be mixed and ingested (Figure 5). After completely emptying the powder, this dispenser is washed and left to dry. It is a very convenient dosing variant in conditions where a scale is not available.

## 4. Discussion

The design of this object was made using modern 3D printing technology having the possibility of creating prototypes that very faithfully imitate a real product. A graduated dispenser was designed for the transportation of the necessary substitute powder for children (aged 6 months–1 year). It has the role of facilitating measurement and transportation, as the designed caps prevent the flow of the powder, and it is able to dose 12 g of the substitute, which means 7.2 g of protein (100 g powder of substitute contains 60 g proteins). It is also practical and very easy to use. It is noted that this substitute dispenser is designed in this way to dose 7.2 g of protein and to transport it to other places where it can be administrated. This device can also be designed to dose and transport a larger amount of PKU substitutes. We designed it only as a prototype for 12 g because our printer does not allow larger projects. (This was the maximum size that could be printed in our CAD Lab.)

**Author Contributions:** Conceptualization, C.A.D.; Data curation M.B.; Methodology, E.P. and C.G.; software, C.G.; resources, E.P., M.B.; writing—original draft preparation C.A.D.; writing—review and editing, C.A.D.; Project administration, C.A.D. All authors have read and agreed to the published version of the manuscript.

**Funding:** This research received no external funding.

**Institutional Review Board Statement:** Not applicable.

**Informed Consent Statement:** Not applicable.

**Conflicts of Interest:** The authors declare no conflict of interest.

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
