# Peer review of "Design and Modelling a Graduated Dispenser for Metabolic Diseases—Phenylketonuria"

_applsci, doi:10.3390/app122010672_

Round 1

Reviewer 1 Report

The authors of this study designed a dispenser for amino acids administered to patients with phenylketonuria. This dispenser is very easy to carry and very useful to patients.

In Material and Methods,  and Results, the authors described a dispenser that is made using modern technology, 3D printing.

The modeling of this dispenser for amino acids is presented in detail in the Figures.

Minor Comment:

The discussion should be extended.

Check all manuscript for grammar errors.

Reviewer 2 Report

The article is very interesting and unique, however, I believe better suited for a journal with a scope on medical parts and engineering designs. I also suggest applying for a patent as this may be a more appropriate means of displaying an invention of this kind. To be suited for a scientific journal, several designs would need to be tested on patients/children to create statistical inference on the impact of this device. In other words, take this experiment a step forward and add a live trial of the device for future consideration in a scientific journal.

1) Also, why use FDM and not SLA or other 3D printing techniques to print the device? FDM may be the most common, but other methods will print a much smoother and robust prototype.

Reviewer 3 Report

Line 45: should clarify the information on when the diet is started, at the time of diagnosis when Phe is equal to or above 360 ​​uM, but during follow-up control of Phe intake is performed when it is above 360 ​​uM in children under 10 years of age and over 600 uM in the elderly (according to European protocol). Since it only mentions in the diagnosis and it is confusing.

Line  48: you must modify the phrase: .. If dietary intervention is not taken in the first 20

days of life, .... stating that the treatment should start as early as possible, preferably before 20 days of age....

Line 54: reformulate sentence..., since the dose of amino acid formula does not depend on tolerance to it but on the protein requirements of the PKU subject.

Line 57: here you should put that the amino acid formula has been fortified with tyrosine, but it does not always cover the requirements of this amino acid, especially in adolescents and adults

Line 75. State what are special foods...foods low in proteins, industrialized?.

Line 79. must be placed because working memory alterations occur in PKU with a neonatal diagnosis since it has been associated with low adherence to treatment.

Line 89: you know that the amino acid formula must be weighed but to facilitate its use, it is recommended to dose it with standardized measures (scoops). And perhaps that would be the goal of your investigation

Line 193: says that the evidence suggests that high doses of formula improve tolerance, this phrase should be reformulated saying that a higher intake of formula increases satiety and thus favors better adherence to treatment and avoids transgressions.

Line 206: They should comment on whether this instrument had a test of its use in the community to evaluate its applicability, primarily if it is intended to be used in schools or by people who do not carry out the treatment, such as teachers or staff in charge of students.

Line 210: when it says that the formula provides 60 g of protein/100 g of powder, you must specify that it is a formula for PKU over 6 years of age since in children under 2 years of age 15 to 30 g/100 g of powder are used. dust.

Reviewer 4 Report

Thank you for inviting me to review the manuscript on a graduated dispenser for metabolic formulas.

As a metabolism provider, I appreciate efforts to help our patients with thoughtful ideas on managing their treatment regimen.

I question the amount of formula you can fit in the dispenser. The maximum amount it fits is a 7.6g protein dose. That is less than the amount required per portion for most of our patients other than the youngest. Also the grams of protein are not the same per volume for every formula. Perhaps for infant formulas, but people with pku are on formulas for their whole lives.

I question how this is superior to using measuring cups/scoops or a gram scale. 

It would seem difficult to get formula by a funnel into a device with this shape. Has it been tried with formula? By patients/family members? I also would like to know if the pressure with pushing the plunger would aerosolize some of the powder resulting in loss of powder or a bit of a mess. Is there something about the size and shape that would make it superior to bottles of Tupperware type containers often used in practice to portion it out?

Just fyi, many people will make 24 hours worth of formula into a big jug and divide it up throughout the day rather than mixing each portion.

Specific suggestions would be expanding on the discussion section and including information about application of this device. The wording is awkward such as the end of the second paragraph when you are talking about dried blood on filter paper testing or dried blood spots (DBS).

The second paragraph on page 2 has the wrong conversion- should be 2-6mg/dl.

You mention hormonal dosing but this is not something in metabolic formulas. 

You could use the term “medical food” where you said “food categories.”

You say in one paragraph people with pku are given 40% about RDA for protein but in the next paragraph you say allude to them gettint 85% plus 15% of the protein requirement, suggesting they are aiming for the RDA. On page 2 on the 3rd full paragraph- Please differentiate between medical food which are protein supplements versus specially formulated low protein foods.

On page 4, I believe you mean 12g x 0.6 = 7.2g

Round 2

Reviewer 2 Report

Authors have addressed previous comments.